# AURKA controls oocyte spindle assembly checkpoint and chromosome alignment by HEC1 phosphorylation

Cecilia S Blengini[1,2] , Shuang Tang[1,2] , Robert J Mendola[3], G John Garrisi[3], Jason E Swain[4], Karen Schindler[1,2]

In human oocytes, meiosis I is error-prone, causing early miscarriages and developmental disorders. The Aurora protein kinases are key regulators of chromosome segregation in mitosis and meiosis, and their dysfunction is associated with aneuploidy. Oocytes express three Aurora kinase (AURK) proteins, but only AURKA is necessary and sufficient to support oocyte meiosis in mice. However, the unique molecular contributions to ensuring high egg quality of AURKA remain unclear. Here, using a combination of genetic and pharmacological approaches, we evaluated how AURKA phosphorylation regulates outer kinetochore function during oocyte meiosis. We found that the outer kinetochore protein Ndc80/HEC1 is constitutively phosphorylated at multiple residues by Aurora kinases during meiosis I, but that serine 69 is specifically phosphorylated by AURKA in mouse and human oocytes. We further show that serine 69 phosphorylation contributes to spindle assembly checkpoint activation and chromosome alignment during meiosis I. These results provide a fundamental mechanistic understanding of how AURKA regulates meiosis and kinetochore function to ensure meiosis I fidelity.

## Introduction

Creation of haploid gametes, eggs and sperm, through meiosis is essential for species propagation. It is therefore critical that meiosis occurs accurately, yet meiosis I is highly error-prone in human oocytes. These errors lead to early miscarriages and developmental disorders (Hassold et al, 2007; Gruhn et al, 2019). Understanding how meiosis I is regulated is therefore critical for determining why human eggs are uniquely prone to chromosome segregation errors.

The Aurora protein kinases are key regulators of chromosome segregation in mitosis and meiosis (Carmena & Earnshaw, 2003; Nguyen & Schindler, 2017). Unlike mitotically dividing cells that express two Aurora kinases (AURKA and AURKB), oocytes express three (AURKA, AURKB, and AURKC). Because it is meiosis-specific, much work has focused on determining AURKC functions in oocytes

(Yang et al, 2010, 2015; Schindler et al, 2012; Balboula & Schindler, 2014; Fellmeth et al, 2015; Balboula et al, 2016; Quartuccio et al, 2017; Vallot et al, 2018; Cairo et al, 2023). However, our works show that AURKA is the only AURK that is necessary and sufficient for mouse oocyte meiosis (Schindler et al, 2012; Nguyen et al, 2018; Blengini et al, 2021a, 2024; Blengini & Schindler, 2024). In determining the spatiotemporal functions of AURKA, we observed that neither targeting AURKA to the acentriolar microtubule–organizing centers (aMTOCs) at spindle poles nor targeting it to the chromosomes fully rescued the spindle and chromosome alignment defects in oocytes lacking all three *Aurk*s (Blengini et al, 2024). We speculated that an additional population of AURKA, either at kinetochores or on spindle microtubules, contributes to spindle elongation and chromosome alignment.

Kinetochore–microtubule (KT-MT) attachments are critical for chromosome alignment and spindle structure. The kinetochore is a complex protein structure that is the interface between centromeres and spindle MTs, regulating MT attachment, MT dynamics, and chromosome movement. The NDC80 complex is part of the outer kinetochore, and it is the direct link between KTs and MTs (Cheeseman et al, 2006; DeLuca & Musacchio, 2012). The NDC80 complex is composed of four proteins: SPC24 and SPC25, which interface with the inner kinetochore, and HEC1 and NUF2, which bind MTs (Musacchio & Desai, 2017). HEC1 undergoes precise phospho-regulation to ensure KT-MT dynamics allowing cycles of attachment and detachment until KT-MT stabilization is achieved (DeLuca et al, 2006). The HEC1 N terminus contains nine phosphorylation sites (Ser4, Ser5, Ser8, Ser15, Ser44, Thr49, Ser55, Ser62, and Ser69) (DeLuca et al, 2006, 2011; Santaguida & Musacchio, 2009). These sites are important for regulating KT-MT affinity, thereby controlling correction of erroneously attached MTs, and they are critical for controlling chromosome oscillation and movement during mitosis (DeLuca et al, 2011; Zaytsev et al, 2014, 2015). In mitosis, AURKA and AURKB phosphorylate HEC1, but the specific contribution of each AURK differs. For example, inhibition of AURKA and AURKB in RPE-1 and HeLa cells showed that Ser55 is phosphorylated by both kinases (DeLuca et al, 2011, 2018; Iemura et al, 2021), whereas Ser44 is primarily phosphorylated by AURKB (DeLuca et al, 2011, 2018), and Ser69 is primarily phosphorylated by AURKA (DeLuca et al, 2018). Based on the pattern of phosphorylation and

[1]Department of Genetics; Rutgers, The State University of New Jersey, Piscataway, NJ, USA   [2]Human Genetics Institute of New Jersey, Piscataway, NJ, USA   [3]CCRM / Institute for Reproductive Medicine and Science (IRMS), Livingston, NJ, USA   [4]CCRM Fertility, Lone Tree, CO, USA

Correspondence: Ks804@hginj.rutgers.edu

the phenotype when each phosphorylation site was altered, different functions for each site were assigned. Although the function of pSer44 is not well understood, pSer55 is critical for destabilization of KT-MT attachments during early prometaphase (DeLuca et al, 2011, 2018) and for chromosome oscillation at metaphase (Iemura et al, 2021). pSer69 is also important for chromosome oscillation at metaphase decreasing the probability of lagging chromosomes (DeLuca et al, 2018; Iemura et al, 2021). In mouse oocytes, HEC1 phosphorylation is important for chromosome alignment (Yoshida et al, 2015), for determining the length of the meiosis I spindle and for restricting aMTOCs at spindle poles (Gui & Homer, 2013; Courtois et al, 2021). However, it is not known how HEC1 phosphorylation is regulated, and which Aurora kinase is involved in fine-tuning this regulation during mouse oocyte meiosis.

To understand the regulatory landscape of HEC1 during oocyte meiosis, we used a combination of genetic and pharmacological approaches to determine the pattern of HEC1 phosphorylation. Furthermore, because oocytes express AURKA, AURKB, and AURKC, we determined the role of each AURK in phosphorylating HEC1 residues. Compared with mitosis, we find differences in meiosis where full phosphorylation of Ser44 requires other unknown kinases, whereas phosphorylation of Ser55 and Ser69 is AURKB/C- and AURKA-specific, respectively. Because AURKA is essential for oocyte meiosis, we focused on determining the role of Ser69 phosphorylation and discovered that its phosphorylation is not only conserved in human oocytes, but also important for regulating spindle assembly checkpoint (SAC) activation and chromosome alignment in mouse oocytes. These results add a new layer of mechanistic understanding of the essential roles of AURKA in regulating kinetochore phosphorylation in oocyte meiosis.

# Results

### Serines 44, 55, and 69 of HEC1 are constitutively phosphorylated during oocyte meiotic maturation

In mitosis, the N terminus of HEC1 is highly phosphorylated during prometaphase and this phosphorylation decreases as the cells approach metaphase. However, it is not known how HEC1 phosphorylation behaves during meiotic maturation in mammalian oocytes. Of the nine HEC1 phosphorylation sites, antibodies that specifically detect five sites (serines 8, 15, 44, 55, and 69) exist. Therefore, we first evaluated which sites are phosphorylated in mouse oocytes. We detected phosphorylation at three serine residues: 44, 55, and 69 (Fig 1A); phosphorylation of serines 8 and 15 was not detectable above the background signal. Next, to determine the temporal pattern of HEC1 phosphorylation during meiotic maturation, we obtained WT mouse oocytes at early prometaphase I, late prometaphase I, and metaphase I and detected these three phosphorylated residues. Although we found a modest, but statistically significant, reduction in pSer55 at metaphase I, the three sites remained constitutively phosphorylated throughout these phases of meiotic maturation and dephosphorylation did not occur at metaphase I (Fig 1B and C). Moreover, we did not detect phosphorylation of Ser8 or Ser15 when tested at earlier meiotic stages (data not shown).

These data suggest that HEC1 phosphorylation and regulation are different in mouse oocyte meiosis I compared with mitosis where they are dephosphorylated as the cell cycle progresses. There are differences between mitosis and meiosis I that could explain these different temporal patterns. In mitosis, the tension between sister chromatids promotes spatial separation of AURKB from its KT substrates. This separation is coordinated with the recruitment of phosphatases to KTs to stabilize the KT-MT attachments via HEC1 dephosphorylation (Foley et al, 2011; Lampson & Cheeseman, 2011; Kabeche & Compton, 2013). Because of the unique meiosis I bivalent chromosome geometry where sister chromatids are attached to one another and not under tension, this separation does not occur even when bivalents are stretched. Therefore, HEC1 and other AURKC KT substrates remain phosphorylated until localized PP2A-B56 phosphatase activity rises enough to counteract AURKC activity (Yoshida et al, 2015). Thus, the intra-KT stretching phase and stabilization of KT-MT attachments are temporally separated (Davydenko et al, 2013; Yoshida et al, 2015; Kitajima, 2018). It is possible that constitutive HEC1 phosphorylation is advantageous, allowing oocytes more time to correct abnormal MT attachments. Alternatively, KT-MT activities may occur with an intermediate level of attachment stability to allow attachment but high turnover. These models require further evaluation.

### Serines 44, 55, and 69 are phosphorylated by Aurora kinases in mouse oocytes

Next, we determined the requirement of the Aurora kinases in HEC1 phosphorylation in mouse oocytes. To address this question, we evaluated the levels of HEC1 phosphorylation at serines 44, 55, and 69 by immunocytochemistry in metaphase I oocytes lacking all three AURKs (ABC-KO). Confirmation of this knockout mouse strain was reported previously (Blengini et al, 2024). We observed that pSer55 and pSer69 were nearly absent in ABC-KO oocytes compared with WT oocytes (Fig 2C–F), suggesting that at least one AURK is critical for the phosphorylation of these residues. However, pSer44 was only reduced by ~50% in ABC-KO oocytes compared with WT oocytes (Fig 2A and B). These data indicate that another kinase also phosphorylates Ser44. We note that the total HEC1 protein was not altered in ABC-KO oocytes, and therefore, decreases in phosphorylation cannot be explained by altered kinetochore structure or protein abundance (Fig S1A and B).

### Aurora kinase specificity for HEC1 phosphorylation sites

Next, to evaluate AURK specificity in phosphorylating these residues in WT oocytes, we used a pharmacological approach applying MLN8237 (MLN) and AZD1152 (AZD) small molecule inhibitors to WT oocytes at concentrations with demonstrated specificity for AURKA or AURKB/C, respectively (Blengini et al, 2022). These experiments were performed at metaphase I to acutely inhibit the kinases. We chose to inhibit the kinases because of documented compensatory mechanisms when the AURKs are deleted or overexpressed (Nguyen et al, 2018; Aboelenain & Schindler, 2021; Blengini et al, 2021b, 2024; Blengini & Schindler, 2024).

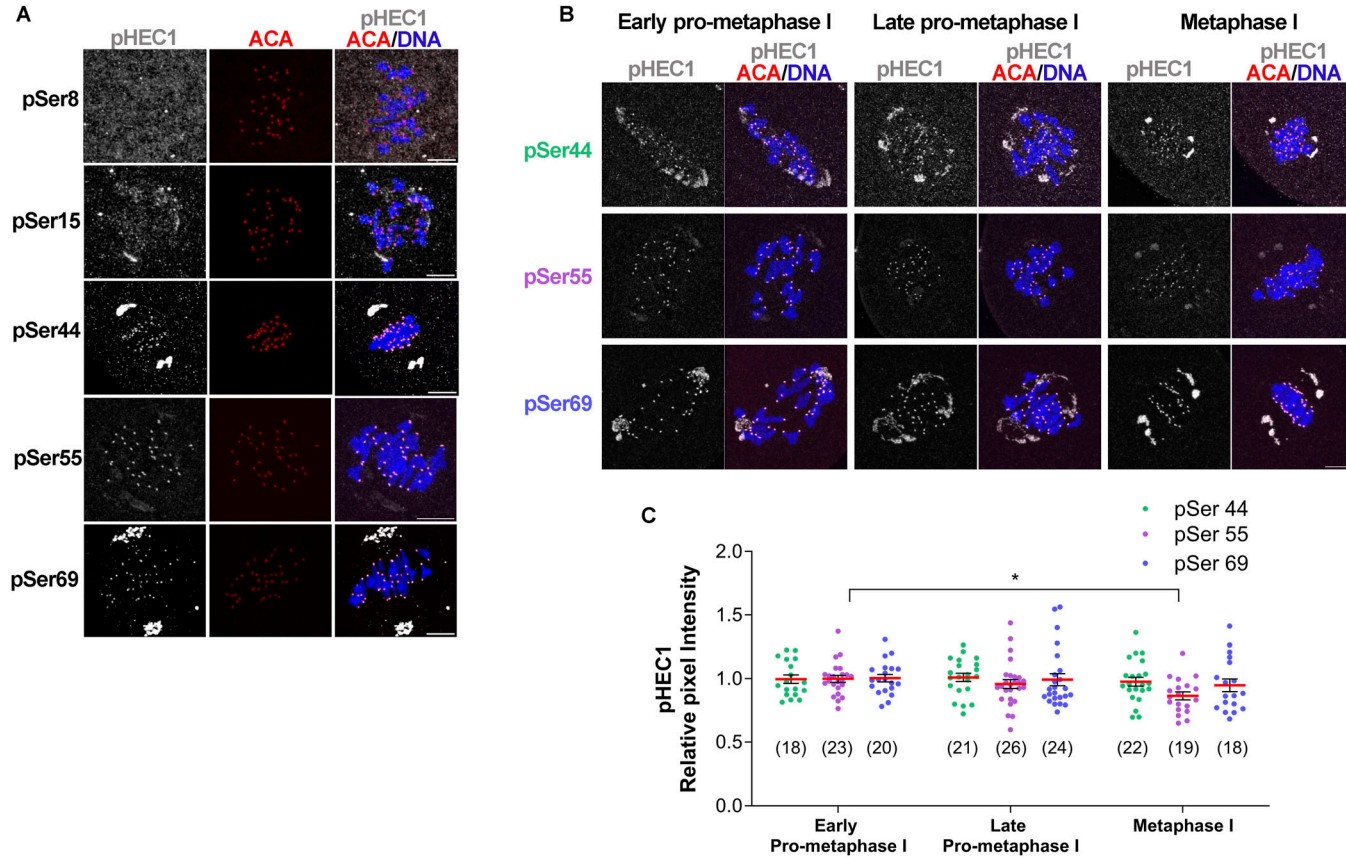

**Figure 1. Temporal pattern of HEC1 N-terminal phosphorylation.**
**(A)** Representative confocal images of WT oocytes stained with phospho-specific antibodies to detect phosphorylated serines (pSer) 8, 15, 44, 55, and 69 (gray), ACA to label kinetochores (red), and DAPI to detect DNA (blue). **(B)** Representative confocal images of WT oocytes matured to early prometaphase I, late prometaphase I, and metaphase I that were stained with phospho-specific antibodies to detect pSer44, pSer55, and pSer69 (gray), ACA to label kinetochores (red), and DAPI to detect DNA (blue). Scale bar: 10 $\mu m$. **(B, C)** Quantification of phosphorylated HEC1 at Ser44 (green dots), Ser55 (purple dots), and Ser69 (blue dots) in (B). The numbers of oocytes analyzed are in brackets from two independent experiments. One-way ANOVA, pSer44: $P = 0.7698$; pSer55: *$P < 0.05$; pSer69: $P = 0.6585$.

### Serine 44

When the inhibitors were applied to WT oocytes, pSer44 was reduced only when AURKB/C were inhibited with AZD and no additional reduction was observed upon inhibition of the three kinases with AZD + MLN (Fig 2G and H). These results suggest that pSer44 is primarily an AURKB/C and not an AURKA target in WT oocytes.

### Serine 55

Upon AURKA inhibition, the pSer55 signal was reduced by 25%, whereas upon AURKB/C inhibition, the pSer55 signal was reduced by 60% (Fig 2I and J). Inhibition of AURKA/B/C did not significantly reduce the pSer55 signal more than AURKB/C inhibition alone. Taken together, the data suggest that serine 55 is primarily phosphorylated by AURKB/C in WT mouse oocytes.

### Serine 69

pSer69 was absent upon inhibition of AURKA with MLN, and no additional reduction was observed when all three kinases were inhibited (Fig 2K and L), suggesting that, as in somatic cells, Ser69 is primarily phosphorylated by AURKA.

### Aurora kinase C can partially compensate for the loss of AURKA in Ser69 phosphorylation

We then asked which AURKs could compensate for one another in phosphorylating HEC1 (Nguyen et al, 2018; Aboelenain & Schindler, 2021; Blengini et al, 2021b, 2024; Blengini & Schindler, 2024). We used previously validated single and double AURK knockout mouse oocytes (Nguyen et al, 2018; Blengini et al, 2021a, 2024) and evaluated the levels of pSer44, pSer55, and pSer69 at metaphase I. pSer44 was only reduced when AURKA and AURKC were deleted in oocytes together (AC-KO) (Fig S2A and B; Table S1), suggesting a redundancy or compensation in phosphorylating Ser44. pSer55 was significantly reduced by 25% in oocytes lacking *Aurkc* (C-KO and AC-KO) (Fig S2C and D; Table S1). The difference in pSer55 immunoreactivity between C-KO and AC-KO was not significantly different, suggesting that AURKB can partially compensate. pSer69 was strongly reduced in all mouse strains lacking AURKA in oocytes (Fig S2E and F; Table S1), and serine 69 phosphorylation was most significantly reduced in AC-KO oocytes (Fig S2E and F; Table S1). These genetic data suggest that AURKC can partially compensate in A-KO oocytes and phosphorylate Ser69.

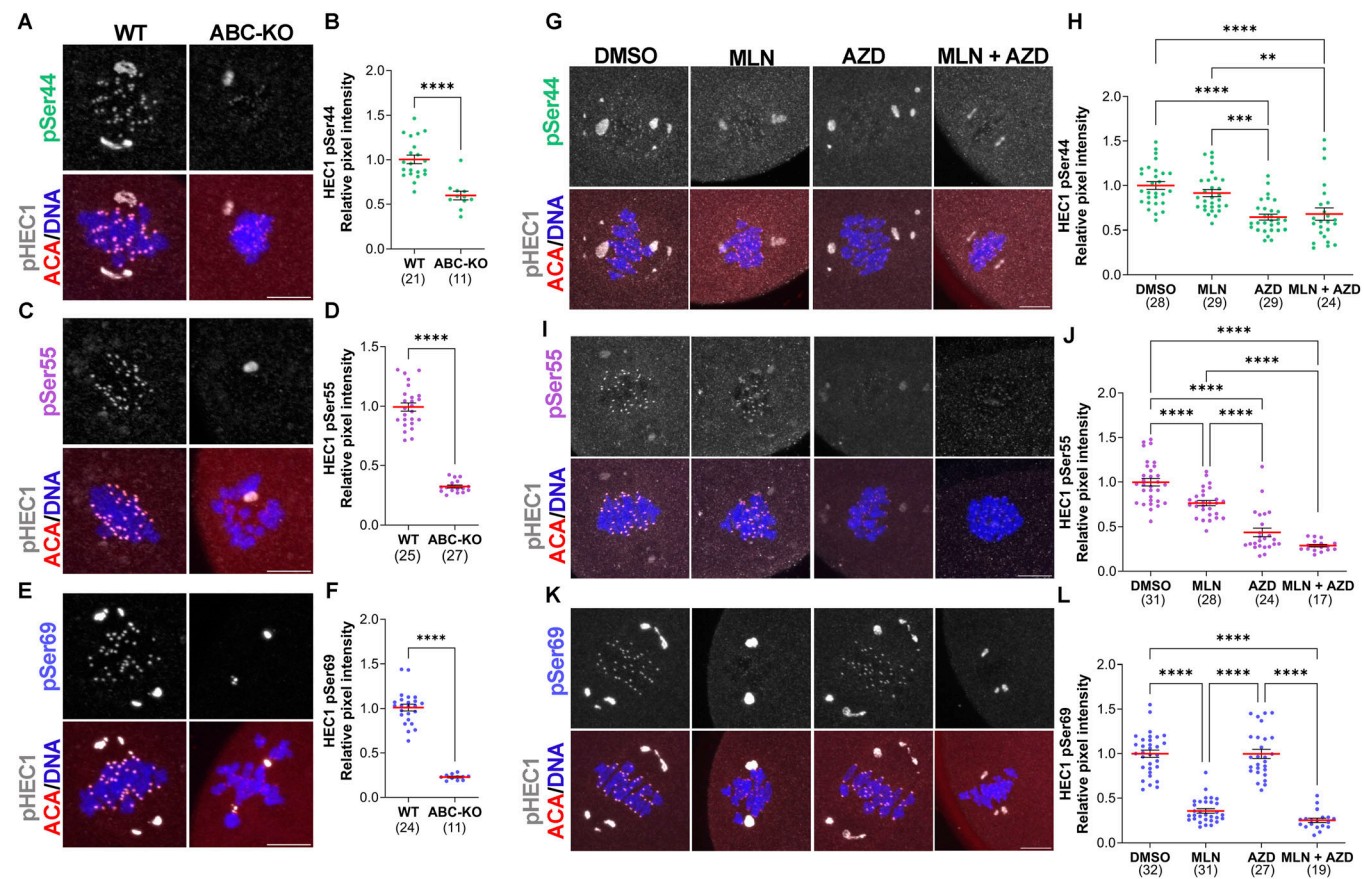

**Figure 2. Specificity of Aurora kinase phosphorylation of HEC1 N-terminal residues.**
**(A, C, E)** Representative confocal images of WT and ABC-KO oocytes at metaphase I stained with phospho-specific antibodies to detect pSer44 (A), pSer55 (C), and pSer69 (E) (gray), ACA (red), and DNA (blue). **(A, B, C, D, E, F)** Quantification of phosphorylated HEC1 at Ser44 (B) (green dots) in (A), Ser55 (D) (purple dots) in (C), and Ser69 (F) (blue dots) in (E). WT oocytes were used as a control. The numbers of oocytes analyzed are in brackets from three independent experiments. *t* test, ****$P < 0.0001$. **(G, I, K)** Representative confocal images of WT oocytes matured to metaphase I and treated with DMSO (control), MLN, AZD, or MLN + AZD and stained with phospho-specific antibodies to detect pSer44 (G), pSer55 (I), and pSer69 (K) (gray), ACA (red), and DNA (blue). **(G, H, I, J, K, L)** Quantification of phosphorylated HEC1 at Ser44 (H) (green dots) in (G), Ser55 (J) (purple dots) in (I), and Ser69 (L) (blue dots) in (K). Oocytes treated with DMSO were used as a control. The numbers of oocytes analyzed are in brackets from two independent experiments. One-way ANOVA, ****$P < 0.0001$. Scale bar: 10 $\mu$m.

These results revealed some similarities and differences in how the N terminus of HEC1 is regulated between mitosis and mammalian oocyte meiosis. For example, Ser44 is less Aurora kinase–specific in meiosis compared with mitosis and less AURKA/B compensation occurs at S55 in meiosis compared with mitosis. However, Ser69 is predominantly phosphorylated by AURKA in both mitosis and oocyte meiosis. Interestingly, AURKB does not appear to primarily phosphorylate any of these sites in oocytes (Fig 2, Table S1), likely because the expression of AURKC, the *Aurk* homolog with the highest homology to *Aurkb*, predominates at kinetochores. However, we cannot completely rule out that AURKB does not phosphorylate HEC1 based on the approaches used here. The expression of the third AURK member promotes a division of tasks during oocyte meiotic maturation, induced by changes in localization and expression levels of each AURK isoform that could affect the pattern of HEC1 phosphorylation. However, AURKA in oocytes localizes similar to somatic cells because AURKA is localized predominantly to spindle poles and has a minor chromosomal population (Kratka et al, 2022). Therefore, AURKA-

dependent phosphorylation of Ser69 in oocytes is conserved between cell types.

### HEC1 Ser69 is phosphorylated in an AURKA-dependent manner in human oocytes

Given the essential functions of AURKA in mouse oocytes, we further investigated whether HEC1 Ser69 is phosphorylated in human oocytes. To answer this question, we obtained discarded, cryopreserved prophase I–arrested human oocytes and matured them in vitro with the MLN and AZD AURK inhibitors; DMSO treatment served as a control. First, Ser69 phosphorylation was detectable in human oocytes in meiosis I, and this mark was localized at kinetochores (Fig 3A). Second, we asked if AURKA is responsible for Ser69 phosphorylation in human oocytes as it is in mouse. We found that pSer69 was significantly reduced upon AURKA inhibition (MLN) and reduced, but not significantly, upon AURKB/C inhibition (AZD) (Fig 3A and B). These results suggest that phosphorylation of HEC1-Ser69 by AURKA is conserved between mouse and human

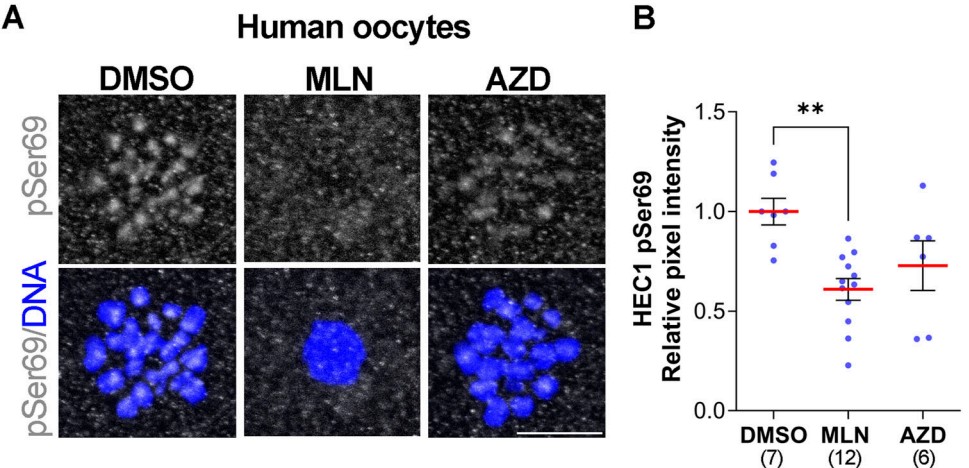

**Figure 3. AURKA phosphorylates HEC1-Ser69 in human oocytes.**
**(A)** Representative confocal images of human oocytes matured and treated with DMSO (control), MLN, or AZD, and stained with a phospho-specific antibody to detect pSer69 (gray). DNA was detected with DAPI (blue). **(B)** Quantification of pSer69 (blue dots) in (B). The numbers of oocytes analyzed are in brackets from three independent experiments. One-way ANOVA, **$P < 0.01$. Scale bar: 10 μm.

oocytes, but that there may be some redundancy with AURKB/C in human that we do not observe in mouse.

### Phosphorylation of HEC1-Ser69 is important for SAC activation

Given the importance of AURKA in oocyte biology and the conservation of HEC1-Ser69 phosphorylation between mouse and human oocytes, we decided to further investigate the role of pSer69 during mouse oocyte meiotic maturation. Serine 55 phosphorylation has been extensively studied elsewhere (Yoshida et al, 2020), and a function has not yet been ascribed to serine 44 phosphorylation. We created HEC1 constructs that encoded either a non-phosphorylated HEC1 variant where Ser69 was mutated to an alanine (69A) or a phospho-memetic HEC1 variant where Ser69 was mutated to an aspartic acid (69D). RNA from these constructs was injected into WT mouse oocytes, which were then cultured for the time it takes control oocytes to reach metaphase II. We found that most oocytes expressing either WT HEC1 or the 69D variant arrested in metaphase I and failed to reach metaphase II. In contrast, 20% of the oocytes expressing the 69A variant progressed past metaphase I and reached metaphase II (Fig 4A and B). As a control, we observed that the expression levels of GFP fusion proteins were comparable between the different constructs (Fig 4C and D), indicating that the phenotypic differences were not because of different expression levels. These results suggest that Ser69 phosphorylation contributes to cell-cycle regulation in oocytes.

Previous studies evaluating somatic cells overexpressing HEC1 showed that these cells arrested in metaphase because of SAC activation and had high levels of MAD2 at kinetochores, a marker of SAC activation (Diaz-Rodríguez et al, 2008; Kemmler et al, 2009). Furthermore, AURKA reportedly can activate the SAC in somatic cells (Courtheoux et al, 2018). To investigate the possibility that Ser69 phosphorylation is involved in regulating SAC activation, we evaluated MAD2 levels at kinetochores in metaphase I oocytes expressing the HEC1 variants. Oocytes with reduced MAD2 recruitment have a weakened SAC and will extrude polar bodies when challenged with SAC-inducing conditions (Wassmann et al, 2003; Homer et al, 2005). Oocytes expressing HEC1-S69A had a 20% reduction in kinetochore-localized MAD2, suggesting that

phosphorylation of Ser69 contributes to SAC activation in mouse oocytes (Fig 4E and F). A caveat with our experimental approach is that the WT oocytes still express endogenous HEC1. Therefore, endogenous WT HEC1 could contribute to the observed mild phenotype in the S69A-expressing oocytes. We note that this result is consistent with previous studies where MAD2 levels were reduced in HEC1-KO oocytes (Yoshida et al, 2020). The results are also consistent with the phenotype where HEC1-KO oocytes express a mutant of HEC1 where nine N-terminal phosphorylation sites were mutated to alanines. Those oocytes underwent premature anaphase I onset, a phenotype consistent with the SAC being prematurely satisfied or weakened (Courtois et al, 2021). Our data extend this finding to show that of the nine phosphorylation sites, phosphorylation of HEC1-Ser69 by AURKA contributes to SAC activation.

### Phosphorylation of HEC1-Ser69 is important for chromosome alignment at metaphase I

HEC1 is also important for promoting spindle bipolarization (Yoshida et al, 2020; Courtois et al, 2021) by recruiting PRC1, an antiparallel MT cross-linking protein, to kinetochores (Yoshida et al, 2020). HEC1 phosphorylation also restricts the aMTOCs to spindle poles (Courtois et al, 2021) and helps ensure chromosome alignment (Yoshida et al, 2015). In our previous study where we analyzed the localized functions of AURKA on meiotic spindle formation, we observed that neither tethering AURKA to aMTOCs nor tethering it to the chromosomes fully rescued the spindle and chromosome alignment defects in ABC-KO oocytes (Blengini et al, 2024). We speculated that we were missing a population of AURKA at kinetochores that contributes to spindle elongation and chromosome alignment. Therefore, we explored whether the AURKA-dependent phosphorylation at Ser69 could improve phenotypic rescue in ABC-KO oocytes. To this end, we overexpressed MTOC-targeted AURKA together with HEC1-S69D in ABC-KO oocytes (Fig 4G) and examined metaphase I spindle volume, spindle bipolarity, and chromosome alignment. We compared these parameters with ABC-KO oocytes expressing either WT-AURKA, MTOC-AURKA, or HEC1-S69D alone as controls.

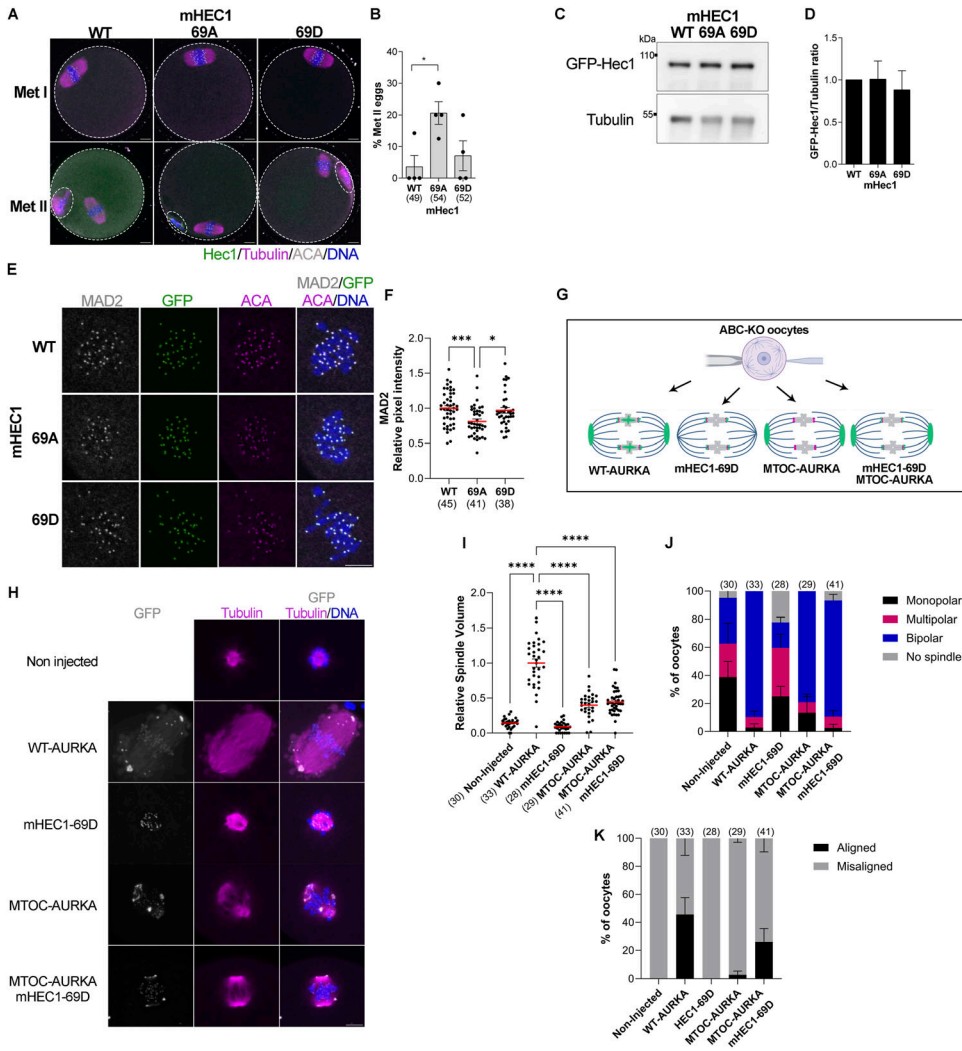

**Figure 4. HEC1-Ser69 phosphorylation regulates SAC activation and chromosome alignment.**
**(A)** Representative confocal images of WT oocytes expressing the indicated *Hec1* mRNAs, matured to metaphase II, and stained with tubulin to detect the spindle (pink), and DAPI to detect DNA (blue). Localization of HEC1-GFPs is in green. The dotted line highlights the oocyte membrane and polar body. **(A, B)** Quantification of the percentage of oocytes at metaphase II in (A). WT oocytes expressing WT-HEC1 were used as a control. One-way ANOVA, *$P < 0.05$ compared with oocytes expressing WT-HEC1. The numbers of oocytes analyzed are in brackets from four independent experiments. **(C)** Representative Western blot images of WT metaphase I oocytes expressing the indicated *Hec1-Gfp* mRNAs. **(C, D)** Quantification of GFP/tubulin ratio from (C). WT oocytes expressing WT-HEC1 were used as a control. One-way ANOVA, $P = 0.8665$ from two independent experiments. **(E)** Representative confocal images of WT oocytes expressing the indicated *Hec1-Gfp* mRNAs at metaphase I and stained with MAD2 (gray), ACA to detect kinetochores (pink), and DNA (blue). Localization of HEC1-GFPs is in green. **(E, F)** Quantification of relative MAD2 pixel intensity from (E). WT oocytes expressing WT-HEC1 were used as a control. One-way ANOVA, ***$P < 0.001$; *$P < 0.05$. The numbers of oocytes analyzed are in brackets from three independent experiments. **(G)** Schematic of the experimental design. ABC-KO oocytes expressing different AURKA-targeted and HEC1-69D constructs are shown in green. Part of this schematic was generated using BioRender. **(H)** Representative confocal images of metaphase I ABC-KO oocytes expressing the indicated mRNAs. Non-injected ABC-KO oocytes were used as a control. Oocytes were stained with tubulin to detect the spindle (green) and DAPI to detect DNA (blue). Localization of the targeted AURKA and HEC1 proteins is in gray. ABC-KO oocytes expressing WT-HEC1 were used as a control. **(H, I)** Quantification of the relative spindle volume from (H). One-way ANOVA, ****$P < 0.0001$ comparing all treatments among each other. The numbers of oocytes analyzed are in brackets from three independent experiments. **(H, J)** Quantification of the percentage of oocytes with different spindle phenotypes in (H). Two-way ANOVA interaction factors: spindle phenotype and oocytes expressing the indicated *Hec1-Gfp* mRNAs, ****$P < 0.0001$. **(H, K)** Quantification of the percentage of oocytes with chromosome misaligned in (H). Two-way ANOVA interaction, ****$P < 0.0001$. Scale bar: 10 μm. In brackets is the number of oocytes analyzed.

First, we evaluated whether the overexpression of these constructs could restore phosphorylation of HEC1 at Ser55 and Ser69. Compared with non-injected control ABC-KO oocytes, the expression of WT-AURKA, MTOC-targeted AURKA, and mHEC1-S69D either alone or in combination did not restore pSer55 in ABC-KO oocytes (Fig S3A). This failure to restore pSer55 is consistent with a requirement for AURKB/C to target Ser55 in WT oocytes (Fig 2I and J). In contrast, pSer69 was restored in ABC-KO oocytes expressing either WT-AURKA or MTOC-targeted AURKA, although to different extents (Fig S3B–E). These results are consistent with our genetic and pharmacological results, showing that Ser69 is primarily phosphorylated by AURKA in mouse oocytes (Fig 2K and L). Moreover, the data show that AURKA localized at aMTOCs can phosphorylate Ser69 at kinetochores, although not to the same extent as WT-AURKA (Fig S3B and C). These results are consistent with AURKA ability to phosphorylate kinetochore substrates when

chromosomes are close to spindle poles in early prometaphase I in oocytes (Chmatal et al, 2015). Alternatively, in oocytes expressing MTOC-targeted AURKA, we observed a dim GFP signal at kinetochores (Blengini et al, 2024), which could be phosphorylating Ser69. The restoration levels of oocytes expressing MTOC-AURKA and HEC1-69D together were reduced compared with oocytes expressing WT-AURKA or MTOC-targeted AURKA alone, suggesting that the expression of HEC1-S69D does not fully replace endogenous HEC1 at kinetochores in ABC-KO oocytes (Fig S3B and E).

We next asked whether spindle parameters and chromosome alignment defects were rescued by co-expressing MTOC-targeted AURKA and HEC1-S69D. ABC-KO oocytes have small spindle volumes, 25% bipolarity, and 100% chromosome misalignment (Fig 4H–K). Consistent with our previous findings, ABC-KO oocytes expressing WT-AURKA had restored metaphase I spindle volumes, 90% spindle bipolarity, and 50% chromosome alignment. When

**AURKA regulation of HEC1 phosphorylation**

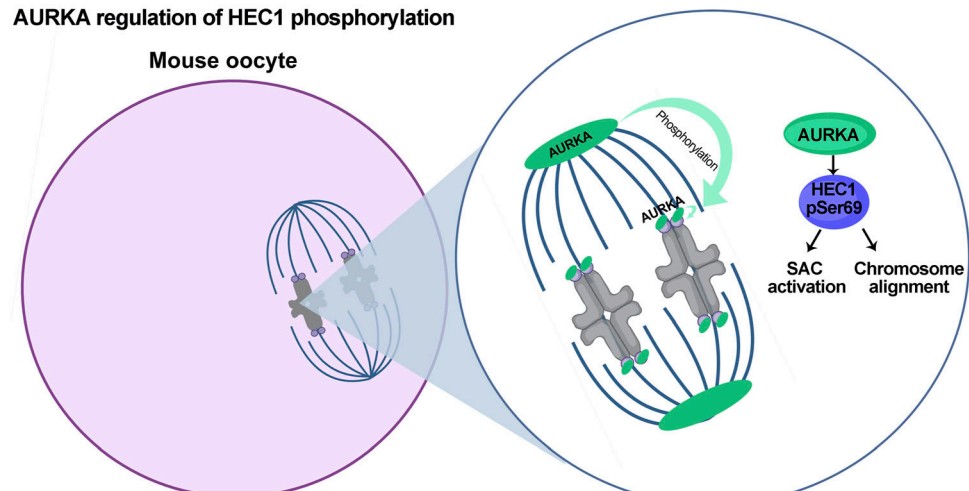

**Figure 5. Model of Aurora kinase A regulation of HEC1 N-terminal phosphorylation in oocytes.**
Populations of AURKA from the spindle poles and the kinetochores regulate spindle assembly checkpoint activation and chromosome alignment during oocyte maturation by the phosphorylation of kinetochore protein HEC1 at Ser69 during oocyte meiosis. Part of this figure was generated using BioRender.

HEC1-S69D was expressed, the phenotype was similar to non-injected control ABC-KO oocytes: small spindle volumes, spindles with mixed polarity, and 100% chromosome misalignment (Fig 4H–K). When MTOC-targeted AURKA was expressed, spindle volumes were partially restored, but, although 80% of the spindles were bipolar, nearly all of the oocytes still had chromosome misalignment (98%) (Fig 4H–K). When MTOC-AURKA and HEC1-S69D were expressed together, spindle volumes and bipolarity were similar to oocytes expressing MTOC-AURKA alone. Interestingly, the percentage of oocytes with chromosome alignment (~30%) was comparable to oocytes expressing WT-AURKA (50%) (Fig 4H–K), suggesting that Ser69 phosphorylation regulates chromosome alignment in mouse oocytes. Taken together, these results suggest that oocytes require AURKA activity at spindle poles and at kinetochores during meiotic maturation. However, whether there are only two discrete populations of AURKA or whether a gradient of AURKA diffuses through the spindle via the liquid-like spindle domain needs further investigation (So et al, 2019).

In sum, we uncovered precise, fundamental mechanisms of how the three AURKs contribute to the phospho-regulation of the HEC1 N terminus in mouse oocyte meiosis. By probing the requirements of Ser69 phosphorylation, we uncovered a new molecular mechanism explaining how AURKA regulates cell-cycle progression by activating the SAC and controlling chromosome alignment during oocyte maturation (Fig 5). Because Ser69 is also phosphorylated in an AURKA-dependent manner in human oocytes, it will be important to investigate whether this mechanism is conserved between mouse and human.

## Materials and Methods

### Mouse strains

To determine the temporal pattern of HEC1 phosphorylation sites and for microinjection experiments, we used WT CF1 females (Envigo) aged 6–12 wk. The knockout strains used in this work were on a C56BL/6N background: wild-type (WT) $Aurka^{f/f}$, and $Aurkb^{f/f}$ but lacking the Cre recombinase transgene; $Aurka^{f/f}$ Gdf9-Cre (A-KO) (Blengini et al, 2021a); $Aurkb^{f/f}$, and Gdf9-Cre (B-KO) (Nguyen et al, 2018); $Aurkc^{-/-}$ (C-KO) (Kimmins et al, 2007; Schindler et al, 2012); $Aurka^{f/f}$, $Aurkb^{f/f}$, and Gdf9-Cre (AB-KO) (Blengini et al, 2024), $Aurka^{f/f}$ $Aurkc^{-/-}$ and Gdf9-Cre (AC-KO); $Aurka^{f/f}$, $Aurkb^{f/f}$ $Aurkc^{-/-}$ and Gdf9-Cre (ABC-KO) (Blengini et al, 2024; Blengini & Schindler, 2024). All mice were described and validated previously. Mice were housed on a 12/12-h light–dark cycle, with constant temperature and with food and water provided ad libitum. Animals were maintained in accordance with guidelines of the Institutional Animal Use and Care Committee of Rutgers University (protocol 201702497). All oocyte experiments were conducted using healthy female mice aged 6–16 wk. Genotyping was performed before weaning and repeated when the animals were used for experiments as previously described (Blengini et al, 2024).

### Mouse oocyte collection, maturation, and microinjection

Prophase I–arrested oocytes were isolated from ovaries of females hormonally primed 48 h earlier with 5 I.U. of pregnant mare's serum gonadotropin (#493–10; Lee Biosolutions). Ovaries were placed in Minimal Essential Medium containing 2.5 $\mu$M milrinone (#M4659; Sigma-Aldrich) to prevent meiotic resumption. To induce meiotic resumption, oocytes were cultured in milrinone-free Chatot, Ziomek, and Bavister (CZB) medium in an atmosphere of 5% $CO_2$ in air at 37°C. To determine the temporal pattern of HEC1 phosphorylation, oocytes were matured for different periods depending on the experimental conditions: to reach early prometaphase I, 3 h after milrinone wash; for late prometaphase I, 5 h after milrinone wash; and for metaphase I, 7–7.5 h after milrinone wash. For acute inhibition of Aurora kinases, oocytes were matured for 7 h to reach metaphase I and then incubated in the respective inhibitor treatment for 3 h. To avoid the entrance to anaphase I, we inhibited the proteasome by adding 5 mM MG132 (#S2619; Selleck Chemicals) to the culture media.

For microinjection, prophase-arrested oocytes were maintained in CZB supplemented with 2.5 µM milrinone to keep them arrested. CF1 oocytes were microinjected with 200 ng/µl mouse HEC1-Gfp (*WT HEC1*), 200 ng/µl mouse HEC1-S69A-Gfp (*HEC1-S69A*), and 200 ng/µl mouse HEC1-69D-Gfp (*HEC1 69D*). ABC-KO oocytes were injected with 100 ng/µl *Aurka-EYfp* (*WT-AURKA*), 100 ng/µl CDK5FRAP fr-*Aurka-EYfp* (*MTOC-AURKA*) (Blengini et al, 2024), and 100 ng/µl mouse HEC1-69D-Gfp (*HEC1 69D*), or with 100 ng/µl CDK5FRAP fr-*Aurka-EYfp* and 100 ng/µl mouse HEC1-69D-Gfp together.

Microinjected oocytes were cultured overnight in CZB supplemented with 2.5 µM milrinone to allow protein expression before the procedures. Subsequently, the oocytes were matured for 7–7.5 h to reach metaphase I.

### Human oocytes

Human oocytes used in this study were discarded immature oocytes (prophase I) originating from patients undergoing routine IVF and elective oocyte cryopreservation cycles under the IRMS/CCRM-NJ IRB-approved protocol (WCG Aspire protocol #20193402). All IRB consent forms include patient consent to participate and publish. All human oocytes were obtained from 11 patients ranging from ages 28 to 37 yr and cryopreserved using vitrification.

Cryopreserved human prophase I oocytes were thawed in pre-warmed serial solutions of 1 M sucrose in M-199 Hepes-buffered medium thawing solution (TS) for 60 s and then 0.5 M sucrose in M-199 Hepes-buffered medium dilution solution (DS) for 3 min. Then, oocytes were recovered in M-199 Hepes-buffered medium washing solution (WS) for 10 min. The oocytes were then moved to pre-equilibrated G-IVF Plus medium (#10136; Vitrolife) supplemented with an additional 10% SPS protein (#ART-3011; SAGE) and incubated at 6.5% $CO_2$ in air at 37°C for 3 h. The oocytes were then moved to pre-equilibrated G-IVF Plus culture drops for continued culture at 6.5% $CO_2$ in air at 37°C.

Oocytes were incubated in pre-equilibrated G-IVF Plus media with the respective inhibitor treatment for 15 h. Oocytes were fixed in 4% PFA in PHEM buffer 1X supplemented with 0.25% Triton X-100 (#1001124827; Sigma-Aldrich) for 30 min at room temperature. The oocytes were transferred to permeabilization solution (PBS containing 0.25% [vol/vol] Triton X-100) for 15 min at room temperature. Oocytes were washed three times with PBS containing 0.05% Tween-20 and blocked in 0.3% BSA containing 0.05% Tween in PBS for 1 h at room temperature. After blocking, the oocytes were incubated in primary antibody in a dark and humidified chamber overnight at 4°C, followed by three washes in washing solution for 10 min each; then, oocytes were incubated in secondary antibody in a dark and humidified chamber for 1 h at 37°C, followed by three washes in washing solution for 10 min each. Lastly, oocytes were mounted in Vectashield (#H-1000; Vector Laboratories) supplemented with 4',6-diamidino-2-phenylindole dihydrochloride (DAPI; #D1306; 1:170; Life Technologies).

### Plasmid information

Preparation of pYX-AURKA-EYFP, pYX-AURKA-EYFP-CDK5RAP2-MBD plasmids was described previously (Blengini et al, 2024). The mouse GFP-HEC1 WT plasmid was a gift from Dr. Iain Cheeseman (MIT/Whitehead Institute), and mHEC1 cDNA (Imageclone 3709641) was amplified by PCR using the following primers: 5′-GCGCGTCTA-GAATGAAGCGCAGTTCAGTTTCCAC-3′ and 5′-TGCTGCCGCGGCATTTGTC GGGAGCCTTAAGTTG-3′ in a backbone with T7 and Xenopus globin UTRs. We mutated the serine 69 to alanine (69A) or aspartic acid (69D) using the QuikChange II Site-Directed Mutagenesis kit (#200523; Agilent Technologies). The primer for mutating S69-A was 5′-TAGCGGACATGGATCCAGGAAT**GCT**CAACTTGGTATATTTTCC-3′, and the primer for mutating S69-D was 5′-CTAGCGGACATGGATCCAGGA AT**GAT**CAACTTGGTATATTTTCCA-3′.

For in vitro transcription, we linearized the plasmids using the Nde1 restriction enzyme and used an mMESSAGE mMACHINE T7 kit to generate RNA (#AM1344; Invitrogen). All mRNAs were purified using Sera-Mag SpeedBead (#GE65152105050250; Sigma-Aldrich) and stored at −80°C.

### Immunofluorescence

Oocytes were fixed in 2% PFA in PHEM buffer 1X for 20 min at room temperature. Immunofluorescence was performed as previously described (Blengini & Schindler, 2018). Briefly, the oocytes were transferred to permeabilization solution (PBS containing 0.1% [vol/vol] Triton X-100 and 0.3% [wt/vol] BSA) for 20 min at room temperature and blocked in 0.3% BSA containing 0.01% Tween in PBS for 10 min. After blocking, the oocytes were incubated in primary antibody in a dark and humidified chamber for 2 h at room temperature, followed by three washes in blocking solution for 10 min each; then, oocytes were incubated in secondary antibody in a dark and humidified chamber for 1 h at room temperature, followed by three washes in blocking solution for 10 min each. Lastly, oocytes were mounted in Vectashield (#H-1000; Vector Laboratories) supplemented with 4',6-diamidino-2-phenylindole dihydrochloride (DAPI; #D1306; 1:170; Life Technologies).

### Western blotting

50 metaphase I oocytes were lysed with Laemmli buffer (#161-0737; Bio-Rad) and denatured at 95°C for 10 min. Proteins were separated by electrophoresis in 10% SDS–polyacrylamide precast gels (#456-1036; Bio-Rad). The separated polypeptides were transferred to nitrocellulose membrane (#170-4156; Bio-Rad) using a Trans-Blot Turbo Transfer System (Bio-Rad) and then blocked with 5% ECL blocking (#RPN418; Amersham) solution in TBS-T (Tris-buffered saline with 0.1% Tween-20) for at least 1 h. The membrane was incubated overnight with anti-GFP (1:500; rabbit; #G1544; Sigma-Aldrich), or 1 h with anti-tubulin (1:1,000; rabbit; 11H10; Cell Signaling Technology) as a loading control. After washing with TBS-T five times, the membranes were incubated with anti-rabbit secondary antibody (1:1,000, #R1006; Kindle Bioscience) for 1 h followed by washing with TBS-T five times. The signals were detected using the ECL Select Western blotting detection reagents (#R1002; Kindle Bioscience) following the manufacturer's protocol. Membranes were stripped before loading control detection using Blot Stripping

Buffer (#46430; Thermo Fisher Scientific) for 30 min at room temperature.

### Antibodies and drugs

To inhibit the Aurora kinases, we used small molecule inhibitors: MLN8237 (AURKA) (alisertib, #S1133; Selleckchem) at 1 $\mu$M and AZD1152 (AURKB/C) (#S1147; Selleckchem) at 100 nM based on previous studies (Blengini et al, 2022). Primary antibodies used were as follows: phosphorylated HEC1 at serine 44 (pSer44) and phosphorylated HEC1 at serine 69 (pSer69) (DeLuca et al, 2011, 2018) (1: 750; rabbit; gifts from Dr. Jennifer De Luca, Colorado State U.); phosphorylated HEC1 at serine 55 (pHEC1 s55) (1:100; rabbit; #GTX70017; GeneTex); anti-HEC1 (1:100; rabbit; a gift from Dr. Robert Benezra, Weill Cornell Medical College); ACA (1:30, #15-234; Antibodies Incorporated); TACC3 (1:100; rabbit; # NBP2-67671; Novus Biologicals); and alpha-tubulin (1:100; sheep; #ATN02; Cytoskeleton). Secondary antibodies were used at 1:200 for IF experiments: anti-mouse Alexa 568 (#A10037; Life Technologies), anti-rabbit Alexa 647 (#A31573; Life Technologies), and anti-sheep Alexa 488 (#A11015; Life Technologies).

### Microscopy

Images were captured using a Leica SP8 confocal microscope equipped with a 40X 1.30 NA oil immersion objective. Optical Z-stacks were obtained using a 0.5-$\mu$m step with a zoom of 4.5. In those experiments where pixel intensity was compared, the laser power was kept constant among genotypes or treatments.

### Image analysis

For pixel intensity at kinetochores in mouse and human oocytes, images were analyzed using ImageJ software (NIH). We created maximum z-projections of the z-sections of each oocyte. To measure the kinetochore pixel intensity of phosphorylated HEC1, ACA was used as a mask to define the region of interest. The threshold algorithm was set in WT or control oocytes. At least 20 individual kinetochores per oocyte were measured, and the average intensity for each oocyte was calculated for these 20 measurements. Relative pixel intensity was determined by dividing the average intensity by the average intensity of all WT/control oocytes in the experiment. For spindle volume measurements, we determined a region of interest based on the tubulin signal at metaphase I using Imaris software (Bitplane) as previously described (Blengini et al, 2024). Chromosome misalignment was defined when at least one chromosome was completely separated from the metaphase plate.

### Quantification and statistical analysis

All experiments were conducted two to three times, any exception would be clarified in the figure legend. t test, one-way ANOVA, or two-way ANOVA was used to evaluate significant differences between/among groups, using Prism software (GraphPad Software). Data are shown as the mean ± SEM.

## Supplementary Information

## Acknowledgements

This work was funded by NIH grant R35 GM136340 to K Schindler. The authors acknowledge Dr. Jennifer DeLuca, Dr. Robert Benezra, and Dr. Iain Cheeseman for generously shared reagents and feedback on the project and manuscript. The authors acknowledge members of the Schindler laboratory for helpful discussions.

### Author Contributions

CS Blengini: conceptualization, investigation, methodology, and writing—original draft, review, and editing.
S Tang: data curation.
RJ Mendola: methodology and writing—review and editing.
GJ Garrisi: resources, methodology, and writing—review and editing.
JE Swain: resources, methodology, and writing—review and editing.
K Schindler: conceptualization, supervision, funding acquisition, and writing—original draft, review, and editing.

### Conflict of Interest Statement

The authors declare that they have no conflict of interest.

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
