## [Reviewer comments · Life Science Alliance]

Life Science Alliance

AURKA controls oocyte spindle assembly checkpoint and chromosome alignment by HEC1 phosphorylation

Cecilia Blengini, Shuang Tang, Robert Mendola, G. John Garrisi, Jason Swain, and Karen Schindler

DOI: <https://doi.org/10.26508/lsa.202403146>

Corresponding author(s): Karen Schindler, Rutgers, The State University of New Jersey

Review Timeline:

Submission Date:	2024-11-25
Editorial Decision:	2025-01-02
Revision Received:	2025-03-25
Editorial Decision:	2025-04-14
Revision Received:	2025-04-23
Accepted:	2025-04-24

Scientific Editor: Tim Fessenden

Transaction Report:

January 2, 2025

Re: Life Science Alliance manuscript #LSA-2024-03146-T

Dr. Karen Schindler
Rutgers, The State University of New Jersey
Genetics
145 Bevier Rd
Piscataway, NJ 08854

Dear Dr. Schindler,

Thank you for submitting your manuscript entitled "AURKA controls oocyte spindle assembly checkpoint and chromosome alignment by HEC1 phosphorylation" to Life Science Alliance. The manuscript was assessed by expert reviewers, whose comments are appended to this letter. We invite you to submit a revised manuscript addressing the Reviewer comments.

Thank you for this interesting contribution to Life Science Alliance. We are looking forward to receiving your revised manuscript.

Sincerely,

B. MANUSCRIPT ORGANIZATION AND FORMATTING:

Reviewer #1 (Comments to the Authors (Required)):

In this study, the authors investigate the role of Aurora kinases (AURKs), specifically AURKA, in regulating the phosphorylation of the kinetochore protein HEC1 during oocyte meiosis, with a focus on the functional consequences of HEC1 phosphorylation at Serine 69 (Ser69). The study provides evidence that HEC1 is constitutively phosphorylated during meiosis I with Ser69 specifically targeted by AURKA in both mouse and human oocytes. Additionally, the authors claim that AURKA-mediated phosphorylation of Ser69 is crucial for spindle assembly checkpoint activation and accurate chromosome alignment during meiosis. The authors propose a conserved mechanism between mouse and human oocytes, highlighting the importance of AURKA-mediated phosphorylation of Ser69 in ensuring successful meiosis I. Overall, by combining genetic and pharmacological approaches the manuscript presents some convincing data that clarifies the role of AURKs in mouse oocytes. The study is well-designed, and the manuscript is in general well-written. However, some further clarifications, including several additional experiments and controls are required prior to the publication of this study.

Major points:

1. The authors quantified HEC1 phosphorylation in different meiotic phases (Fig. 1B-C), showing a significant reduction in Ser55 phosphorylation in metaphase I. However, the selected images do not optimally represent the stated meiotic phases (i.e. late pro-metaphase I does not look different than early pro-metaphase I and the cell selected for pSer44 in metaphase I appears to be in late pro-metaphase I), which rises concerns about the accuracy of the quantification of pHEC1 levels during meiosis.
2. For pHEC1 quantification (and any other pixel intensity quantification done in the manuscript) the authors use maximum z-projections. Sum or average z-projections should be used to correctly assess the fluorescence signal in the samples. Also, ACA signal should be used to normalize the quantification of pHEC1 signal at kinetochores (pHEC1/ACA).
3. In the text, the authors hypothesize that the difference between the phosphorylation status of HEC1 in meiosis and mitosis could be due to the mitosis-specific spatial separation of AURKB from its kinetochore substrates (which could also be true for AURKC). However, phosphorylation of Ser69 (phosphorylated by AURKA) is maintained, whereas phosphorylation of Ser55 (phosphorylated by AURKC) is reduced in Fig. 1C. How do the authors explain this difference?
4. The authors show that phosphorylation at Ser55 site of HEC1 is highly reduced in the presence of AURKB/C inhibitor (~60% reduction in Fig. 2H). They conclude that Ser55 is primarily phosphorylated by AURKC. Can AURKB activity be ruled out here?
5. AURKB/C inhibition reduced pSer55 signal by 60% (Fig. 2 G-H) compared to AURK C-KO, which reduced pSer55 signal only by 25% (Fig. S1A-B), and AURK BC-KO, which did not reduce pSer55 signal at all (Fig. S1A-B). How the authors explain these differences given their conclusion that Ser55 is primarily phosphorylated by AURKC? Moreover, pSer55 reduction in AC-KO condition is more significant than in C-KO alone. This suggests that AURKA partially compensates to phosphorylate Ser55 when AURKB and AURKC are absent.
6. Total HEC1 levels should be presented and quantified too. If total HEC1 is affected by ABC-KO, the quantification of pHEC1 status should be normalized to total HEC1.
7. The authors state that AURKB does not phosphorylate any of the studied HEC1 sites (lines 167-169). However, the AZD inhibitor used in the study is not specific to AURKC, and any experiments that could rule out the role of AURKB in pSer55 are missing. Moreover, although it is known that this residue is mainly phosphorylated by AURKB during mitosis (DeLuca, 2018), experiments assessing pSer44 status in the presence of AURK inhibitors are also missing. Thus, this claim is not fully supported by the experimental work and should be rephrased or supported with new experimental data.
8. Although the graph in Fig. 2M shows only around 40% reduction in pSer69 signal in human oocytes, its corresponding representative image in Fig. 2K shows a complete absence of pSer69 signal. It is not clear how pSer69 quantification was performed in Fig. 2M. ACA or other kinetochore marker that can be used as a mask to define the ROI for pHEC1 quantification should be used and presented.
9. The authors state that phosphorylation of Ser69 in HEC1 by AURKA is responsible for SAC activation (lines 220-222). The experiments presented in Fig. 3E-F show reduced MAD2 levels upon overexpression of a phosphonull version of pSer69 HEC1. However, the authors do not directly relate AURKA activity with SAC activation in this context. To strengthen their claims, the authors could assess MAD2 levels after AURKA inhibition. The effect of AURKA inhibition on MAD2 levels should be even higher, as endogenous HEC1 (which is not depleted in the overexpression experiments) would also be dephosphorylated.
10. Optimally, Fig. 3H-K should also include non-KO control cells.
11. Lines 277-278: The authors state that "This mechanism is likely conserved between mouse and human.", referring to AURKA and SAC activation. However, the only experimental data in human oocytes shows that AURKA mainly phosphorylates pSer69 HEC1. Thus, this is an overstatement and should be rephrased.

Minor points:

1. Images in Fig. 1A display different meiotic phases for different pSer detection. Since HEC1 phosphorylation status could vary through the cycle (as it does in mitosis), a more detailed analysis (including early pro-metaphase I, late pro-metaphase I, and metaphase I) could be done. For instance, Fig. 1B could include pSer8 and pSer15, giving a better comparison of different pHEC1 sites.
2. Experiments using AURK inhibitors (Fig. 2) are performed in metaphase I, whereas pSer55 is shown to be less phosphorylated in that phase compared to pro-metaphase I (Fig. 1B). Could this initial reduction affect the results?
3. The KO efficacy for the experiments done in Figures 2 and S1 should be presented.
4. There is a prominent difference in the reduction of phosphorylation levels (especially for pSer55) when comparing ABC-KO with single and double AURKs KO (around 60-70% reduction in ABC-KO (Fig. 2D) vs. 25% in C- and AC-KO (Fig. S1B)). What could be the reason for that?
5. The authors only show the effect of MLN and AZD in pSer55 and pSer69. However, pSer44 levels are also significantly reduced in the ABC-KO experiment (Fig. 2A). It would be interesting to assess how pSer44 behaves in the presence of AURKs inhibitors.
6. The number of human oocytes should be stated in the conditions presented in Fig. 2K.
7. The text states that "Ser69 phosphorylation was detectable in metaphase I human oocytes" in line 181, but the images shown do not look like metaphase I. There is a general discrepancy between the meiotic phases stated in the text and the ones shown in the images (i.e. indicating metaphase I when showing late pro-metaphase I).
8. Why is AURK inhibitor treatment much longer (15 h) in human oocytes compared to mouse oocytes (3 h)? Could this long treatment have other effects on the cells?
9. The total number of experiments is stated as "All experiments were conducted 2-3 times", but the exact number is only stated in Fig. 1. This should be fixed.
10. Scale bars are missing in some images (ideally each individual cell should include its scale bar).
11. The used controls should be better described in the methods section or figure legends.
12. More detail is needed on how spindle volume was quantified.
13. The figure legend should include information on what is indicated in a dotted circle in Fig. 3A.
14. Statistics should be clearer in line 440, and are missing in figure legend for Fig. 3B, as well as in Fig. 3J and 3H (what is being compared?)

Reviewer #2 (Comments to the Authors (Required)):

AURKA controls oocyte spindle assembly checkpoint and chromosome alignment by HEC1 phosphorylation by Cecilia S. Blengini and collaborators.

In this manuscript the authors report that AURKA specifically phosphorylates S69 on HEC1 to maintain/activate the SAC in mouse oocytes while AURKC phosphorylates S55. They also report that these phosphorylation events are conserved in human oocytes.

The data are convincing, but the manuscript would benefit from being restructured. This would make it accessible to a wider audience.

Major comments

COMMENT #1

100 Serines 44, 55 and 69 of HEC1 are constitutively phosphorylated during oocyte meiotic 101 maturation.

In this first paragraph the authors demonstrated that Ser 44, 55 and 69 on HEC1 are phosphorylated during oocyte maturation in early pro-metaphase I, late-pro-metaphase I and metaphase I.

I understand that the authors are interested by HEC1 epitopes phosphorylated at kinetochores, so they presented figures of microscopy but did they perform any Western Blots to evaluate the level of phosphorylation of HEC1?

Saying this, I realize that it might be difficult to use mouse oocytes for WBs.

COMMENT #2

131 Aurora kinases A and C phosphorylate HEC1 at Serine 55 and 69 in mouse oocytes

In the second paragraph the authors show that Ser 55 and 69 are not phosphorylated in the triple AURKs KO oocytes suggesting that they are targeted by AURKs. Then they used AURKs inhibitors MLN8237 (MLN) and AZD1152 (AZD) to discriminate between AURKA and AURKC target.

I don't understand why the authors decided to use inhibitors.

Why not trying to re-express AURKA or AURKC in the triple KO AURKA/B/C?

This would have been cleaner and would have strengthened the manuscript.

Especially since the authors have mastered the technique they use in the last paragraph.

This paragraph should be dedicated to unambiguously identify the kinase that phosphorylate S69

COMMENT #3

189 Phosphorylation of HEC1-Ser69 is important for spindle assembly checkpoint activation
190 and chromosome alignment at metaphase I

This paragraph is too dense, it should be dedicated to identify the function of the phosphorylated sites on HEC1. Phosphorylation of S69 by AURKA and phosphorylation of S55 by AURKC. I would have added experiments by repressing WT-AURKC and HEC1-55D alone or HEC1-55D-69D to eventually prove that the observed phenotypes are really due to those phosphorylations.

COMMENT #4

To summarize: the manuscript should be structured as
Identification of sites phosphorylated by AURKs on HEC1
Identification of which AURK phosphorylates which site
Identification of phenotypes associated to a lack of phosphorylation of each site
Rescue using S/D mutants to prove the mechanism

Minor comments

MINOR COMMENT #1

270 meiotic maturation. However, if there are only two discrete populations of AURKA or if a gradient
271 of AURKA diffuses through the spindle via the liquid-like spindle domain needs further
272 investigation.

I'm surprised not to see the reference of the first paper reporting the role of AURKA kinase in SAC, even if it concerns mitosis. AURKA kinase activity was reported (1) to be required to maintain SAC active in prometaphase (2) to maintain the localization of Mad2 on unattached kinetochores, and (3) was detected localized to the kinetochores. (Courtheoux et al, J Cell Sci. 2018 Apr 12;131(7):jcs191353. doi: 10.1242/jcs.191353)

MINOR COMMENT #2

192 phosphorylation between mouse and human oocytes, we decided to further to investigate the
193 role of pSer69 during mouse oocyte meiotic maturation.

to further to investigate : remove "to" before investigate

Dear Editor,

We thank you and the reviewers for the careful review and suggestions for improving our manuscript. We have taken their comments into consideration and conducted several experiments to address their concerns. We ask that you consider our current staffing and budgetary limitations in reviewing the revised manuscript. The first author now lives in Texas, is no longer an employee and is on maternity leave. Our funding is in the last few months of the budget and the renewal is held up from NIH freezes (was supposed to go to NIH Council in February). Together, these have created unprecedented obstacles. But we have tried to address as many points as possible. Importantly, we do not find changes in total HEC1 in the triple knockout, supporting the model that there are Aurora kinase-dependent phosphorylation changes and not simply alterations in kinetochore structure. We have also added to the figures and some numbers and panel lettering have changed.

Below are is our point-by-point response to the reviewers' critiques.

Reviewer 1:

1. The authors quantified HEC1 phosphorylation in different meiotic phases (Fig. 1B-C), showing a significant reduction in Ser55 phosphorylation in metaphase I. However, the selected images do not optimally represent the stated meiotic phases (i.e. late pro-metaphase I does not look different than early pro-metaphase I and the cell selected for pSer44 in metaphase I appears to be in late pro-metaphase I), which rises concerns about the accuracy of the quantification of pHEC1 levels during meiosis.

Response: We appreciate the reviewer's keen eye and concern. When conducting these experiments, we fix oocytes at specific time points which represent the indicated cell cycle stage. The early pro-Metaphase I time point was at 3h after induction of meiosis and late pro-Metaphase I at 5h. We selected images from these time points. Early pro-metaphase oocytes vary in phenotype- sometimes chromosomes are very spread apart and sometimes they are more condensed. We now replace the early pro-metaphase images to better represent early and late stages. Because we are rigid in our time point sampling and used many oocytes in our replicates, we are confident that our quantifications are an accurate reflection of the average.

2. For pHEC1 quantification (and any other pixel intensity quantification done in the manuscript) the authors use maximum z-projections. Sum or average z-projections should be used to correctly assess the fluorescence signal in the samples. Also, ACA signal should be used to normalize the quantification of pHEC1 signal at kinetochores (pHEC1/ACA).

Response: We appreciate that different labs have different preferred methods of image analysis. We have published over 40 manuscripts using the maximum projection analysis method and we always normalize to controls. Therefore, we are confident in our assessments presented here, especially because the controls are treated in the same manner and used as reference. But, to ensure that we have not mis-interpreted the data, we re-analyzed phosphorylation of the Ser69 meiotic maturation (Figure 1B)

dataset using the average method and normalizing to ACA (Figure below). We found that changing the method of the analysis did not alter the result or conclusion. Reanalyzing all the data in this suggested method would take a significant amount of time and because the first author is now unemployed and living in Texas and is on maternity leave, this is not feasible and not a great use of the lab's time. We hope that the reviewer is convinced that our method is not only sound but understands our preference to keep the analysis as it is.

3. In the text, the authors hypothesize that the difference between the phosphorylation status of HEC1 in meiosis and mitosis could be due to the mitosis-specific spatial separation of AURKB from its kinetochore substrates (which could also be true for AURKC). However, phosphorylation of Ser69 (phosphorylated by AURKA) is maintained, whereas phosphorylation of Ser55 (phosphorylated by AURKC) is reduced in Fig. 1C. How do the authors explain this difference?

Response: This is a question that we would love to answer but cannot at this time. We speculate that AURKA localization is different from AURKC and that it does not become spatially separated in metaphase. It is possible that AURKC resides on the inner side of the kinetochore that becomes stretched away when the chromosomes are under tension and AURKA resides on the outer side that remains in close contact. Ultimately, we do not know enough about how proteins at the meiotic kinetochore behave to draw a conclusion.

4. The authors show that phosphorylation at Ser55 site of HEC1 is highly reduced in the presence of AURKB/C inhibitor (~60% reduction in Fig. 2H). They conclude that Ser55 is primarily phosphorylated by AURKC. Can AURKB activity be ruled out here?

Response: We apologize for this overinterpretation made because our other studies indicate minor, non-canonical AURKB roles in oocyte meiosis. The reviewer is correct, with the inhibitor result on its own, we cannot rule out AURKB activity. We now modify the text at line 182 to read: "Taken together, the data suggests that Serine 55 is primarily phosphorylated by AURKB/C in WT mouse oocytes." This is now Figure 2I-J.

5a. AURKB/C inhibition reduced pSer55 signal by 60% (Fig. 2 G-H) compared to AURKC-KO, which reduced pSer55 signal only by 25% (Fig. S1A-B), and AURKB-C-KO,

which did not reduce pSer55 signal at all (Fig. S1A-B). How the authors explain these differences given their conclusion that Ser55 is primarily phosphorylated by AURKC?

Response: Our conclusion is based on the inhibitor experiment which is testing the endogenous, WT situation. That is, when B/C are inhibited, Ser55 is reduced compared to when A was inhibited. The smaller difference observed in the KOs reflects partial compensation by AURKA which we reported in Nguyen *et al*, *Current Biology*, 2018.

5b. Moreover, pSer55 reduction in AC-KO condition is more significant than in C-KO alone. This suggests that AURKA partially compensates to phosphorylate Ser55 when AURKB and AURKC are absent.

Response: We agree that there is partial compensation of AURKA in the B/C KO oocyte. We first reported this in a previous publication that fully analyzes the B/C KO phenotype (<https://pubmed.ncbi.nlm.nih.gov/30415701/>). This finding was our motivation for using the genetics to test compensation ability for phosphorylating HEC1 (indicated at Line 190). When taken together, our results indicate that in WT oocytes, Ser55 is primarily phosphorylated by AURKB/C but that AURKA can compensate.

6. Total HEC1 levels should be presented and quantified too. If total HEC1 is affected by ABC-KO, the quantification of pHEC1 status should be normalized to total HEC1.

Response: Thank you for this great suggestion. We have now conducted this experiment and confirm that total HEC1 is not affected by loss of the AURKs. This is now included as a new Figure S1 and described in the revised text at line 153. Because the kinetochore structure is not altered, we do not need to normalize to HEC1. Furthermore, all antibodies were raised in rabbit, so co-staining is not feasible.

7a. The authors state that AURKB does not phosphorylate any of the studied HEC1 sites (lines 167-169). However, the AZD inhibitor used in the study is not specific to AURKC, and any experiments that could rule out the role of AURKB in pSer55 are missing.

Response: Testing AURKB function is tricky in oocytes. Over the years, we and others tried knockdown techniques but they lacked specificity. Overexpression is another approach, but does not represent the WT scenario, so results would not be precise if it increased phosphorylation. As indicated above, we have modified our language to not exclude AURKB in this function.

7b. Moreover, although it is known that this residue is mainly phosphorylated by AURKB during mitosis (DeLuca, 2018), experiments assessing pSer44 status in the presence of AURK inhibitors are also missing. Thus, this claim is not fully supported by the experimental work and should be rephrased or supported with new experimental data.

Response: Thank you for this concern. For a focused report on sites with known functions, we had elected to not include this data on Ser44, but had in fact conducted

this analysis. We have added this data back into the manuscript in new Figure 2G, H and Figure S1A, B. As you will see, we found that Serine 44 is primarily phosphorylated by AURKB/C in WT oocytes, although AURKA can compensate when AURKB/C are absent. Based on our results in ABC-KO, Ser44 is also being phosphorylated by another, unknown kinase that needs further investigation. We now detail this in the text starting at line 169.

8. Although the graph in Fig. 2M shows only around 40% reduction in pSer69 signal in human oocytes, its corresponding representative image in Fig. 2K shows a complete absence of pSer69 signal. It is not clear how pSer69 quantification was performed in Fig. 2M. ACA or other kinetochore marker that can be used as a mask to define the ROI for pHEC1 quantification should be used and presented.

Response: Thank you for the concern about the selected image for Figure 2K. We now present an image that falls in the average for pSer69 signal, rather than at the bottom of the intensity spread. To quantify pHEC1 we did use an ROI of ACA and detail that in the methods. We have modified our figure layout and this is now Figure 3 in the revision.

9. The authors state that phosphorylation of Ser69 in HEC1 by AURKA is responsible for SAC activation (lines 220-222). The experiments presented in Fig. 3E-F show reduced MAD2 levels upon overexpression of a phosphonull version of pSer69 HEC1. However, the authors do not directly relate AURKA activity with SAC activation in this context. To strengthen their claims, the authors could assess MAD2 levels after AURKA inhibition. The effect of AURKA inhibition on MAD2 levels should be even higher, as endogenous HEC1 (which is not depleted in the overexpression experiments) would also be dephosphorylated.

Response: Thank you for this suggestion. We have conducted this experiment by acutely inhibiting AURKA and did not find a change in MAD2 levels (see Figure below). This can be explained by the difference in the experiment- a global inhibition of AURKA and all its pathways vs changing 1 local phospho-site. We know that AURKA is required for multiple events in oocytes including activating translation of maternal mRNAs, a step required to progress past Metaphase I (Aboelenain and Schindler 2021, <https://pubmed.ncbi.nlm.nih.gov/34636397/>) and building a bipolar spindle (Blengini et al. 2024, <https://pubmed.ncbi.nlm.nih.gov/39081293/>). We do not believe this negative result rules out our interpretation, but we now temper our conclusion in the discussion of this data at Line 294.

10. Optimally, Fig. 3H-K should also include non-KO control cells.

Response: We thank you for that consideration. As we were designing the experiment, we did consider this control. However, not only would adding WT control cells double the extremely laborious workflow, but our goal was to assess rescue of the triple KO, not a defect. Performing these studies in WT oocytes would be testing a different question- that is, testing an overexpression phenotype. Therefore, our decision was to not include the WT oocytes as they were not really a control for the experimental question.

11. Lines 277-278: The authors state that "This mechanism is likely conserved between mouse and human.", referring to AURKA and SAC activation. However, the only experimental data in human oocytes shows that AURKA mainly phosphorylates pSer69 HEC1. Thus, this is an overstatement and should be rephrased.

Response: We have rephrased this statement to say: "Because Ser69 is also phosphorylated in an AURKA-dependent manner in human oocytes, it will be important to investigate if this mechanism is conserved between mouse and human." (Line 362)

12. Images in Fig. 1A display different meiotic phases for different pSer detection. Since HEC1 phosphorylation status could vary through the cycle (as it does in mitosis), a more detailed analysis (including early pro-metaphase I, late pro-metaphase I, and

metaphase I) could be done. For instance, Fig. 1B could include pSer8 and pSer15, giving a better comparison of different pHEC1 sites.

Response: Thank you for this suggestion. We now evaluated Ser8 and Ser15 phosphorylation at early pro-Metaphase and find that like our previous result where pSer8 and pSer15 are not detected at kinetochores (Figure below). We used pSer55 as a positive control for our staining. We expand on this in the text at Line 123.

13. Experiments using AURK inhibitors (Fig. 2) are performed in metaphase I, whereas pSer55 is shown to be less phosphorylated in that phase compared to pro-metaphase I (Fig. 1B). Could this initial reduction affect the results?

Response: This timing would not affect the results because we do not compare the Metaphase I signal to pro-Metaphase I. Instead, we compared Metaphase I controls to Metaphase inhibited. To make sure that we are comparing the same stage, we included MG132 in the culture to keep controls and treated oocytes in Metaphase.

14. The KO efficacy for the experiments done in Figures 2 and S1 should be presented.

Response: These KO strains and their efficacy have been previously reported in our published manuscripts. We indicated this in the methods text but realize we should indicate this in the main text. We have now modified this in the revision at lines 150 and 192 and provide the relevant citations.

15. There is a prominent difference in the reduction of phosphorylation levels (especially for pSer55) when comparing ABC-KO with single and double AURKs KO (around 60-70% reduction in ABC-KO (Fig. 2D) vs. 25% in C- and AC-KO (Fig. S1B)). What could be the reason for that?

Response: Thank you for pointing out this difference. We had not considered this in our evaluation. This difference would indicate that AURKB can phosphorylate Ser55 in a double KO genetic background. We now indicate this in the text at line 199.

16. The authors only show the effect of MLN and AZD in pSer55 and pSer69. However, pSer44 levels are also significantly reduced in the ABC-KO experiment (Fig. 2A). It would be interesting to assess how pSer44 behaves in the presence of AURKs inhibitors.

Response: Although S44 was reduced in the triple KO, it was only reduced by 50%, compared to the near absence of the other 2 marks. As we indicate in the text, this result suggests that another kinase targets S44, at least in the absence of the 3 AURKs. For a focused report on sites with known functions, we had elected to not include this data on Ser44, but had in fact conducted this analysis. We have added this data back into the manuscript in Figure 2G-H and Figure S1A, B. As you will see, we found that Serine 44 is primarily phosphorylated by AURKB/C in WT oocytes, although AURKA can partially compensate when AURKB/C are absent and now detail this in the text at line 194.

17. The number of human oocytes should be stated in the conditions presented in Fig. 2K.

Response: We apologize for the omission. We now include these numbers in the graph (now Figure 3).

18. The text states that "Ser69 phosphorylation was detectable in metaphase I human oocytes" in line 181, but the images shown do not look like metaphase I. There is a general discrepancy between the meiotic phases stated in the text and the ones shown in the images (i.e. indicating metaphase I when showing late pro-metaphase I).

Response: Working with cryopreserved human oocytes is tricky. They take a longer time to recover and re-enter meiosis. We classified any oocyte without a polar body at the end of our experiment as metaphase, however, as the reviewer indicates, these may not be truly metaphase. We revised our language to say meiosis I (line 237).

19. Why is AURK inhibitor treatment much longer (15 h) in human oocytes compared to mouse oocytes (3 h)? Could this long treatment have other effects on the cells?

Response: Human oocytes spend a significant longer time in prometaphase compared to mouse (see <https://pubmed.ncbi.nlm.nih.gov/26045437/>). In addition, it takes time for cryopreserved oocytes to restart meiosis, compared to reports of freshly isolated oocytes. Therefore, the treatment required a longer incubation. We do not believe it had other effects on the oocytes because they were still viable and appeared healthy looking. The Aurora B/C inhibition with AZD was conducted for the same amount of time without significant change in S69 phosphorylation, suggesting that the incubation time

with inhibitors was not an issue. But, because these oocytes are in limited supply, these are not assays that we can optimize as we would with mouse so it is impossible to fully rule out. We hope that the reviewer can appreciate the value of using human oocytes in this study despite the limitations.

20. The total number of experiments is stated as "All experiments were conducted 2-3 times", but the exact number is only stated in Fig. 1. This should be fixed.

Response: We apologize for the omission. These replicates are now indicated in all figure legends.

21. Scale bars are missing in some images (ideally each individual cell should include its scale bar).

Response: When we image our experiments, we establish zoom settings that remain constant for the entire acquisition and image all the oocytes in a single setting. Therefore, the scale bars do not change from oocyte to oocyte. The images selected for figure presentation are from the single setting. Therefore, our preference is to keep the single scale bar in each figure panel as we find this style more aesthetically pleasing.

22. The used controls should be better described in the methods section or figure legends.

Response: We apologize for this lack of clarity. We now indicate the controls in all figure legends.

23. More detail is needed on how spindle volume was quantified.

Response: We now add additional information to the methods on how spindle volume was quantified.

24. The figure legend should include information on what is indicated in a dotted circle in Fig. 3A.

Response: The dotted circle follows the oocyte membrane and polar body. We now indicate this in the legend which is now Figure 4.

25. Statistics should be clearer in line 440, and are missing in figure legend for Fig. 3B, as well as in Fig. 3J and 3H (what is being compared?)

Response: We apologize for this lack of clarity. We now indicate the comparisons in the figure legend which is now Figure 4.

Reviewer 2:

1. In this first paragraph the authors demonstrated that Ser 44, 55 and 69 on HEC1 are phosphorylated during oocyte maturation in early pro-metaphase I, late-pro-metaphase I and metaphase I. I understand that the authors are interested by HEC1 epitopes phosphorylated at kinetochores, so they presented figures of microscopy but did they perform any Western Blots to evaluate the level of phosphorylation of HEC1? Saying this, I realize that it might be difficult to use mouse oocytes for WBs.

Response: Thank you for the suggestion to supplement our microscopy data with western blots to examine total levels. As the reviewer indicates, this is difficult for mouse oocytes because of the limiting amounts of material. It is also complicated by the fact that we received the phospho-specific antibodies as a gift and they are in limited supply. In our experience with oocyte western blotting, we have found that it typically takes several rounds of optimization to trouble-shoot how many oocytes are required for obtaining a signal and a working dose and work flow for the antibody. On average, it takes about 100 oocytes to obtain a signal for most antibodies – this is 4 mice, per meiotic stage, per replicate after using many more mice for troubleshooting. Because we are interested in kinetochore-localized signal, we feel that total phospho HEC1 signals would not add significant information to support the use of a large number of mice and depleting our antibody stock. We hope that the reviewer shares this same conclusion.

2. Line 131. I don't understand why the authors decided to use inhibitors. Why not trying to re-express AURKA or AUKC in the triple KO AURKA/B/C? This would have been cleaner and would have strengthened the manuscript. Especially since the authors have mastered the technique they use in the last paragraph. This paragraph should be dedicated to unambiguously identify the kinase that phosphorylate S69.

Response: We decided to use inhibitors because we knew that the KOs and ectopic expression would have compensatory issues. Our goal was to reveal how the WT/endogenous system works. In the revision, we break this section apart to highlight S69 phosphorylation, as suggested and add rationale of the compensation to the text at Line 164.

3. Line 189- “Phosphorylation of HEC1-Ser69 is important for spindle assembly checkpoint activation and chromosome alignment at metaphase I”. This paragraph is too dense, it should be dedicated to identify the function of the phosphorylated sites on HEC1. Phosphorylation of S69 by AURKA and phosphorylation of S55 by AURKC. I would have added experiments by repressing WT-AURKC and HEC1-55D alone or HEC1-55D-69D to eventually prove that the observed phenotypes are really due to those phosphorylations.

Response: We appreciate the advice on improving the readability and have divided up this section to improve focus. Our manuscript was originally submitted to a parent journal in a short report format and then directly transferred to LSA. We therefore had to

compress information to comply with that format. We elected to not follow up on AURKC and S55 because AURKC is not essential for oocyte meiosis and S55 function has been explored by other groups. By contrast, AURKA is essential for oocyte meiosis and S69 phosphorylation has not yet been reported. Therefore, we felt that the more impactful work would focus on essential and new biology. We hope that the reviewer agrees with our logic.

4. To summarize: the manuscript should be structured as Identification of sites phosphorylated by AURKs on HEC1; Identification of which AURK phosphorylates which site; Identification of phenotypes associated to a lack of phosphorylation of each site; Rescue using S/D mutants to prove the mechanism.

Response: We have edited the manuscript to improve flow and readability. As described above, we have elected to remain focused on S69 and not add S55 function into the story.

5. Inclusion of the first paper reporting the role of AURKA kinase in SAC, even if it concerns mitosis. (Courtheoux et al, J Cell Sci. 2018 Apr 12;131(7):jcs191353. doi: 10.1242/jcs.191353)

Response: Thank you for alerting us to this manuscript. We apologize for failing to incorporate this reference and supporting information in our manuscript. We now include this reference as rationale for evaluating the SAC at Line 274.

6. Line 192- "phosphorylation between mouse and human oocytes, we decided to further to investigate the role of pSer69 during mouse oocyte meiotic maturation". remove "to" before investigate

Response: Thank you for catching that typo. It is now corrected.

April 14, 2025

RE: Life Science Alliance Manuscript #LSA-2024-03146-TR

Dr. Cecilia S. Blengini
Rutgers, The State University of New Jersey
Genetics
145 Bevier Rd
Piscataway, NJ 08854-8009

Dear Dr. Blengini,

Thank you for submitting your revised manuscript entitled "AURKA controls oocyte spindle assembly checkpoint and chromosome alignment by HEC1 phosphorylation". As you will see, reviewers are satisfied that their concerns have been resolved. We would be happy to publish your paper in Life Science Alliance pending final revisions necessary to meet our formatting guidelines.

- Please be sure that the authorship listing and order is correct.
- Please add an ORCID ID for the corresponding author- you should have received instructions on how to do so.
- Please add a Summary Blurb/Alternate Abstract and a Category for your manuscript in our system.
- Please add the X and Bluesky handles of your host institute/organization as well as your own or/and one of the authors in our system.
- Please add your main, supplementary figure, and table legends to the main manuscript text after the references section.
- We encourage you to revise the figure legends for figures 4 and S3 such that the figure panels are introduced in alphabetical order.
- Per LSA policy, the contributions indicated for G. John Garrisi and Jason E. Swain do not qualify them for authorship. Please either update the contributions in our system and the Author Contributions section of the manuscript or let us know if the authors need to be removed (and potentially added to the acknowledgment section).
- Please use the [10 author names, et al.] format in your references (i.e., limit the author names to the first 10).
- Please add a Conflict of Interest statement to your main manuscript text.
- There are callouts for figure S1E-F and this figure doesn't have these panels -- please correct. Please add callouts for Figure S2A-F to your main manuscript text.
- Please add weight to the blots in Figure 4C.

A. FINAL FILES:

-- Summary blurb (enter in submission system): A short text summarizing in a single sentence the study (max. 200 characters including spaces). This text is used in conjunction with the titles of papers, hence should be informative and complementary to

the title. It should describe the context and significance of the findings for a general readership; it should be written in the present tense and refer to the work in the third person. Author names should not be mentioned.

B. MANUSCRIPT ORGANIZATION AND FORMATTING:

Sincerely,

Reviewer #1 (Comments to the Authors (Required)):

Most of my concerns have been appropriately addressed in the revised manuscript and thus I support its publication in Life Science Alliance.

Reviewer #2 (Comments to the Authors (Required)):

In this manuscript, the authors analyse the phosphorylation of HEC1 by Aurora kinases during mouse and human oocyte maturation. They found that phosphorylation of Ser69 by AURKA contributes to the spindle assembly checkpoint and chromosome alignment.

I had few comments after reading the first version of the manuscript, the authors are providing me with answers to all my comments. I think the manuscript is ready for publication.

April 24, 2025

RE: Life Science Alliance Manuscript #LSA-2024-03146-TRR

Dr. Karen Schindler
Rutgers, The State University of New Jersey
Genetics
145 Bevier Rd
Piscataway, NJ 8854

Dear Dr. Schindler,

Thank you for submitting your Research Article entitled "AURKA controls oocyte spindle assembly checkpoint and chromosome alignment by HEC1 phosphorylation". It is a pleasure to let you know that your manuscript is now accepted for publication in Life Science Alliance. Congratulations on this interesting work.

DISTRIBUTION OF MATERIALS:

Again, congratulations on a very nice paper. I hope you found the review process to be constructive and are pleased with how the manuscript was handled editorially. We look forward to future exciting submissions from your lab.

Sincerely,
